



# Local evaporation controlled by regional atmospheric circulation in the Altiplano of the Atacama Desert

Felipe Lobos-Roco[1,2], Oscar Hartogensis[1], Jordi Vilà-Guerau de Arellano[1], Alberto de la Fuente[3], Ricardo Muñoz[4], José Rutllant[4,5], and Francisco Suárez[2,6,7]

[1]Meteorology and Air Quality, Wageningen University, Wageningen, The Netherlands.
[2]Department of Hydraulic and Environmental Engineering, Pontificia Universidad Católica de Chile, Santiago Chile.
[3]Department of Civil Engineering, Universidad de Chile, Santiago, Chile.
[4]Department of Geophysics, Universidad de Chile, Santiago, Chile.
[5]Centro de Estudios Avanzados en Zonas Aridas, La Serena, Chile.
[6]Centro de Desarrollo Urbano Sustentable (CEDEUS), Santiago Chile.
[7]Centro de Excelencia en Geotermia de los Andes (CEGA), Santiago Chile.

**Correspondence:** Felipe Lobos Roco (felipe.lobosroco@wur.nl; felipe.lobos.roco@gmail.com)

**Abstract.** We investigate the influence of regional atmospheric circulation on the evaporation of a saline lake in the Altiplano region of the Atacama Desert. For that, we conducted a field experiment in the Salar del Huasco (SDH) basin (135 km east of the Pacific Ocean), in November 2018. The measurements were based on surface energy balance (SEB) stations and airborne observations. Additionally, we simulate the meteorological conditions on a regional scale using the Weather Research and

Forecasting model. Our findings show two evaporation regimes: (1) a morning regime controlled by local conditions, in which SEB is dominated by the ground heat flux ($\sim$0.5 of net radiation), very low evaporation ($L_v E < 30$ W m$^{-2}$) and wind speed $<1$ m s$^{-1}$; and (2) an afternoon regime controlled by regional-scale forcing that leads to a sudden increase in wind speed ($>15$ m s$^{-1}$) and a jump in evaporation to $>500$ W m$^{-2}$. While in the morning evaporation is limited by very low turbulence ($u*$ $\sim$0.1 m s$^{-1}$), in the afternoon strong winds ($u*$ $\sim$0.65 m s$^{-1}$) enhance the mechanical turbulence, increasing the evaporation.

We find that the strong winds in addition to the locally available radiative energy are the principal drivers of evaporation. These winds are the result of a diurnal cyclic circulation between the Pacific Ocean and the Atacama Desert. Finally, we quantify the advection and entrainment of free-tropospheric air masses driven by boundary-layer development. Our research contributes to extend our understanding of evaporation drivers in arid regions and how large-scale processes affect directly local ones.

## 1 Introduction

The Atacama Desert is known as the driest place on Earth, with precipitation ranging from 0.1 mm per decade ($\sim$0.01 mm yr$^{-1}$) in the lowlands (Weischet, 1975) to 150-180 mm yr$^{-1}$ (Minvielle and Garreaud, 2011) in the highlands. The Altiplano (highlands) is rain-fed by occasional convective showers, whose source of humidity arrives from the East (Falvey and Garreaud, 2005). These storms are spatially very localized and rapidly changing in intensity ($<1$ hour), being the sole source of aquifer recharge and thus sustain the shallow lagoons and wetlands that host unique native floral and faunal environments (De

La Fuente and Niño, 2010; Johnson et al., 2010). It is in these confined water-holding environments that nearly all the water





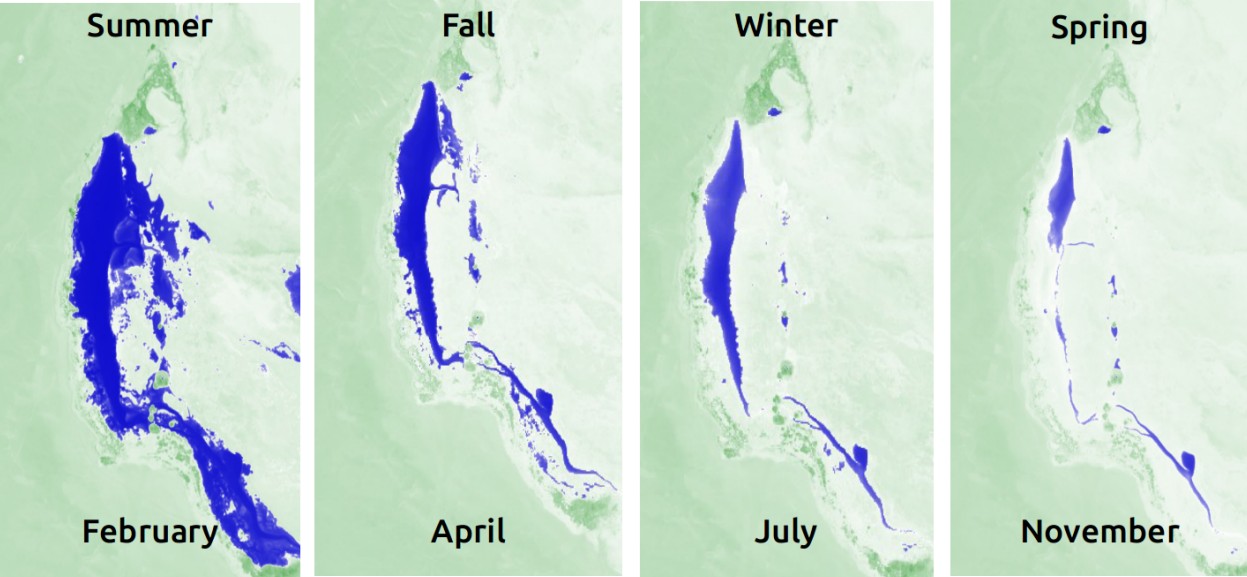

**Figure 1.** Shallow saline lake at Salar del Huasco as viewed by the normalized difference water index (NDWI) from Copernicus Sentinel data 2019 processed by Sentinel Hub. This index combines infrared and visible bands, where dark blue represents water and light green the absence of water. The right-hand image shows the extent of the lake on November $18^{th}$ 2018, during the field measurements shown in this work.

of the catchment is lost to the atmosphere, meaning that they act as a preferential pathway for evaporation (E) (Rosen, 1994). Consequently, locally at these lagoons the annual E greatly exceeds annual precipitation (Lictevout et al., 2013). In this study we focus on a particular saline lake, the Salar del Huasco (SDH) located in the Altiplano of the Atacama Desert in the NE region of Chile. The dynamics of the E-process of SDH can be regarded as exemplifying all the saline lakes in the region (Kampf et al., 2005). Figure 1 shows the dramatic change in the size of the SDH lake from the rainy season in the summer to the dry season in late spring. Between winter (June-September) and spring (September-December) the size of the lagoon is reduced by 75% in only two months. Our study focuses on this part of the year that represents the peak of the annual E water-loss.

The main mechanisms that drives this high E rate are not yet well understood, yet they are crucial to improve its representation in atmospheric and hydrological models and thus to improve water management efficiency. Atmospheric model calculations of E in arid and semi-arid regions are still uncertain for several reasons. First, the physical processes governing E occur on spatial scales smaller than the usual model grid size (~1 km), over heterogeneous surfaces and on sub-hourly temporal scales (Eder et al., 2014). Second, modelled E rates are mainly controlled by net radiation ($R_n$) and water vapour-pressure deficit ($vpd$) (Ma et al., 2018). However, in reality more complex processes take place over arid regions (McNaughton, 1976; De Bruin et al., 2005). More specifically, induced atmospheric flows driven by local surface heterogeneity play an important role





in governing the surface energy balance (SEB) (Moene and Van Dam, 2014) while on a regional scale, advection can enhance E to the point at which it exceeds $R_n$ (De Bruin et al., 2005). This multi-scale interaction between surface and atmosphere impacts the vertical atmospheric boundary layer (ABL) structure, which feeds back into E-related processes at the surface (van Heerwaarden et al., 2010). On larger scales, the meteorological influence at the regional level is particularly relevant in the

Atacama Desert due to the complex topography and thermal gradient between the atmosphere above the Pacific Ocean and the western slope of the Andes mountains that results in an energetic atmospheric flow every afternoon (Rutllant et al., 2003). Under these premises the role of the regional atmospheric circulation and its interaction with surface processes is crucial to an understanding of the E diurnal cycle.

The aim of this study is to describe and quantify the physical processes that control diurnal cycle of E in arid regions char-

acterized by confined water environments and the surfaces of their heterogeneous surroundings over the complex topography of the SDH. More specifically, our main research question is: in the interplay between regional and local scales, what is the role of the wind-induced turbulence in controlling the diurnal cycle of E compared to $R_n$ and $vpd$ as the main drivers? An understanding of this would help to improve representations of E in numerical models and potentially improve the efficiency of water resources management in arid regions.

To unravel what processes and scales control E, we combine observations gathered during a field experiment called E-DATA (**E**vaporation caused by **D**ry **A**ir **T**ransport over the **A**tacama Desert) that took place in the SDH in November 2018. The analysis of the observations is supported by fine-resolution numerical experiments using the Weather Research and Forecasting (WRF) model. The originality of the designed field experiment is that it integrates ground and airborne observations over heterogeneous surfaces to quantify the moisture and energy budgets as well as the interaction between ABL and the regional

circulation. The modeling perspective includes multi-day numerical model runs to quantify the regional flow patterns.

This manuscript is structured as follows. Section 2 presents the basic theoretical concepts utilized in this study. Section 3 describes the methods employed and data gathered in the field experiment and modeling. Section 4 presents the main results, describing surface fluxes and their relationships and interactions with the environmental conditions in local and regional perspectives. Section 5 summarizes the processes involved and discusses the results in the context of other studies. Finally, the

main conclusions and future perspectives are presented.

## 2  Basic conceptual framework of evaporation

Environmental conditions in the Atacama Desert are characterized by abundant radiation (incoming shortwave radiation, $S_i$ max $> 1000$ W m$^{-2}$), dry air (specific humidity, $q < 1$ g kg$^{-1}$), limited soil moisture ($\sim 0$ m$^3$ m$^{-3}$) and in some parts very

low plant transpiration. The main sources of E are the saline lakes in the endorheic (closed) basins, which is controlled by the interplay of energy (radiation), wind (turbulent mixing) and $vpd$ between the confined open surface water and the atmosphere (McNaughton, 1976).

To analyze the relevance of the main processes related to evaporation in our measurements, we employ the concept of the





Penman equation for open-water evaporation (Penman, 1948; Monteith, 1965) expressed in terms of energy, i.e., the latent
heat flux ($L_v E$). This analysis aims to demonstrate the qualitative behavior of the Penman-Monteith steering variables to show
which mechanisms and conditions are limiting E. The equation reads.

$$L_v E = \frac{s}{s+\gamma} \overbrace{(R_n - G)}^{\text{I}} + \frac{\rho_a c_p}{s+\gamma} \overbrace{\underbrace{\frac{1}{r_a}}_{\text{turbulence}} \underbrace{(e_s - e)}_{\text{vpd}}}^{\text{II}}, \tag{1}$$

where $s$ is the slope of saturated vapour pressure curve, $\gamma$ the psychrometric constant, $\rho_a$ is the dry air density and $c_p$ is the
specific heat at constant pressure. We indicate terms in Equation (1) that represent the two main processes that contribute to
$L_v E$. The term I is the $energy$ contribution (Garratt, 1992), which describes the energy available ($R_n$ - $G$) to evaporate water
where $R_n$ is the net radiation and $G$ the ground heat flux. The term II is the $aerodynamic$ contribution, which combines the
turbulence and water vapour pressure deficit ($vpd$) contribution. Here, the first sub-term describes the efficiency of turbulent
mixing, where $r_a$ is the aerodynamic resistance defined as:

$$r_a = \frac{1}{ku^*} \left[ \ln\left(\frac{z}{z_{0,h}}\right) + \Psi\left(\frac{z_{0,h}}{L}\right) - \Psi\left(\frac{z}{L}\right) \right], \tag{2}$$

where $k$ is the von Kármán constant (0.4), $u*$ is the friction velocity, $z$ is the height of measurements and $z_{0,h}$ is the roughness
length for heat, $\Psi$ is the integrated stability function for heat in the entire atmospheric surface layer (Paulson, 1970) and $L$
the Monin-Obukhov length. Note that the usual Penman-Monteith equation term referred to the stomatal resistance is omitted,
due to the absence of vegetation in the study area. Last, the second sub-term on the right-hand side of Equation (1), the $vpd$
contribution, describes the pressure-deficit of the water vapour ($e_s - e$) at the level measured (see Table 1).

The two terms in Equation (1) represent the main drivers for E. The $energy$ contribution (term I) is related to local-scale
conditions prescribed by surface processes (section 4.1) while the $aerodynamic$ contribution (term II) is related to both local
and regional scale interactions (sections 4.2 and 4.3). An important aspect of our research is to quantify the relevance of non-
local effects. Examples of non-local processes are the advection of heat and moisture and the entrainment of air from above the
ABL. Both transports modify the local $vpd$ values and thus influence the diurnal variability of E (De Bruin et al., 2005). These
non-local processes impact ABL development in the entrainment zone, which also influences E rates (van Heerwaarden et al.,
90  2009).

Finally, in order to distinguish local from non-local and regional contributions to the changes in the potential temperature, $\theta$
and specific humidity, $q$ across a boundary layer with height, $h$, we make use of the mixed-layer approximation. Here, our
aim is to determine under which conditions the $\theta$ and $q$-budget follow the mixed-layer approximation (Stull, 1988). In case the
approximations are valid, we can use these equations to quantify the contributions by using the observations. The mixed-layer
equations read:

$$\frac{\partial q}{\partial t} = \frac{\overline{w'q'_s} - \overline{w'q'_e}}{h} - U\frac{\partial q}{\partial x} \tag{3}$$





and

$$\frac{\partial \theta}{\partial t} = \frac{\overline{w'\theta'_s} - \overline{w'\theta'_e}}{h} - U\frac{\partial \theta}{\partial x}, \tag{4}$$

where $t$ is the time, $\overline{w'q'}$ and $\overline{w'\theta'}$ are the kinematic moisture and heat fluxes, sub-indices $s$ and $e$ are for surface and entrainment at the top of the boundary layer respectively, $U$ is the total wind speed and $x$ the spatial direction aligned with the main horizontal wind.

The first term of the right-hand side of equation (3) and (4) represents the local and non-local contributions of the vertical fluxes that are distributed over the boundary layer. In our modeling framework, the surface fluxes $\overline{w'q'_s}$ and $\overline{w'\theta'_s}$ are parameterized as a function of resistance and the gradients between the value at the surface and the mixed-layer value (Vilà-Guerau de Arellano et al., 2015). The second term, represents the transport of air with different properties coming from elsewhere, which we refer to as regional contributions. Typically, the regional contribution is estimated as a residual term from locally measured fluxes and vertical profiles of $\theta$ and $q$ (details in Appendix A3).

## 3 E-DATA experiment: observations and modeling

The E-DATA (**E**vaporation caused by **D**ry **A**ir **T**ransport over the **A**tacama Desert) field experiment consisted of horizontally distributed SEB and meteorological (MET) stations over the SDH saline lake and the heterogeneous surfaces that surround it, and vertical atmospheric measurements (Suárez et al., 2020). The E-DATA experiment was designed to analyse both local (∼1 km) and regional scales (∼100 km). The measurements were complemented with a comprehensive 3D-regional modeling study with the WRF atmospheric meso-scale model. In this section we will provide a site description (3.1), descriptions of the surface observations (3.2), the profiling measurements (3.3) and the WRF modeling set-up (3.4).

### 3.1 Site description and instrumentation set-up

The E-DATA experiment was performed between $14^{th}$ and $23^{rd}$ November 2018 at the SDH (20.1° S - 68.5° W, 3790 m above sea level (asl)). This date is optimal to study evaporation due to the total absence of precipitation and high mean temperatures. The SDH is a closed basin of 1417 km$^2$ (55 km N-S and 35 km W-E) located ∼3.8 km up and over ∼135 km from the Pacific Ocean. Note that at such altitude, the pressure level is very low compared to sea level, ∼650 hPa. Figure 2a shows the location of the SDH saline lake and E-DATA experiment in a vertical cross-section over the western slope of the Andes mountains. Figure 2b shows an overview of the surface observation installation in the vicinity of the SDH saline lake. Three SEB-stations were installed over representative and homogeneous surfaces of the site: water, wet-salt, and desert. The first SEB-station was installed above a shallow 15-cm deep lagoon (20.27° S - 68.88° W; 3790 m asl), whose surface covers 4 km N-S by 800 m W-E. The second SEB-station was located over a wet-salt crust (20.28° S, 68.87° W; 3790 m asl), which is a wet soil composed of salt whose surface is covered by a mostly dry crust of slime. The third SEB-station was installed in an area representative


**Figure 2.** Study site. (a) Vertical cross-section of western slope of the Andes, showing the spatial scales involved in the field experiment and modeling. (b) Spatial distribution of surface and vertical observations at SDH site during the E-DATA field experiment used in this study (Contains modified Copernicus Sentinel data processed by Sentinel Hub). (c) WRF outer domains D01 (27 km), D02 (9 km) and inner domains D03 (3 km), D04 (1 km). The SDH saline lake is located at the center of the D04 and dotted line indicates the vertical cross-section shown in section 4.3.





of bare rocky-soil-like desert conditions (20.35° S, 68.90° W; 3953 m asl). Figure 2b also shows the profiling measurement points from where radiosonde and an unmanned aerial vehicle (UAV) were launched: water and desert. The first launch site was located on the western -shore of the lagoon (20.28° S - 68.88° W; 3790 m asl) covering water and wet-salt surfaces. The second point was located next to the desert SEB-station (20.35° S - 68.90° W; 3953 m asl), covering the desert surface that
surrounds the SDH-basin. A transect of four automatic MET-station deployed from 20.28° S - 68.90° W to 20.28° S - 68.97° W westward of the lagoon was utilized to characterize the advection. Finally, we also made use of a standard meteorological station placed 2 km N from the saline lake (Fig. 2b), that has been in continuous operation since 2015 by the Centro de Estudios Avanzados en Zonas Aridas (CEAZA).

## 3.2  Surface observations


We deployed SEB-stations, complemented by additional meteorological measurements, at each of the three main surface types (water, wet-salt, desert) together with a transect of four MET-stations on the western slopes of the study site. Special attention is paid to the measurement of variables that are related to the drivers of E: radiation, turbulence and $vpd$. Table 1a shows the main variables and sensors utilized over each surface type organized by sensor groups. Radiation measurements and sensors
differed between surfaces. The four-component radiation measurements were gathered for the water surface, whereas at the desert and wet-salt surfaces only integrated $R_n$ measurements were available. The albedo was measured only at the water site, for the wet-salt it was estimated using the net short-wave radiation and the incoming short-wave radiation measured in the lake. For the desert site we assumed the value 0.21, reported as a typical value for dry sandy soils in Moene and Van Dam (2014). Additionally, the $R_n$ of the desert SEB station was too large so it was corrected by using incoming short-wave measurements
from the water SEB-station, and assuming an albedo of 0.21 (Moene and Van Dam, 2014). We used the flux-software package EddyPro v 6.2.2 (Fratini and Mauder, 2014) from LI-COR Biosciences Inc. (Lincoln, Nebraska, USA) to calculate the turbulent fluxes of latent heat ($L_vE$), sensible heat ($H$) and the friction velocity ($u*$). All standard data treatment and flux correction procedures were included, most notably axis rotation with the planar-fit procedure (Wilczak et al., 2001), raw data screening including spike removal (Vickers and Mahrt, 1997), interval linear detrending and low-pass filtering correction (Massman,
2000). In addition, quality flags were determined based on Mauder and Foken (2004). The measured ground heat flux ($G$) was corrected for heat storage above the heat-flux plates by using the calorimetric method (Kimball and Jackson, 1975), and the observations obtained from soil temperature probes buried at different depths (see Table 1) in each surface type. Note that over a shallow water layer G is stored in both the water and soil/sediment layers above the heat flux plates (de la Fuente and Meruane, 2017). We corrected for both components of the soil heat storage. Standard meteorological variables such as air
temperature ($T$), relative humidity ($RH$), atmospheric pressure ($P$), wind speed ($U$) and wind direction ($WD$) were measured in the SEB-stations and at a transect of standard meteorological stations. The details are shown in Table 1a. The uncertainty related to the energy balance closure at the SEB stations can be found in Appendix A.



**Table 1.** Main variables and sensors utilized during the E-DATA experiment, by sensor-group and surfaces: water (W), wet-salt (WS) and desert (D). (a) Surface main variables: incoming shortwave radiation ($S_i$), outgoing shortwave radiation ($S_o$), incoming longwave radiation ($L_i$) and outgoing longwave radiation ($L_o$), shortwave net radiation ($SW_{net}$), longwave net radiation ($LW_{net}$), net radiation ($R_n$), latent heat flux ($L_vE$), sensible heat flux ($H$), friction velocity ($u*$), ground heat flux ($G$), soil temperature ($T_{soil}$), air temperature ($T$), relative humidity ($RH$), wind speed ($U$), wind direction ($WD$) and pressure ($P$). (b) Vertical main variables.

| Sensor-group | Surface | Main-variable | height [m] | Sensors |
|---|---|---|---|---|
| a | | | | |
| | W | $S_i, S_o, L_i, L_o$ | 1 | CNR4 Net radiometer [a] |
| Radiation | WS | $R_n$ | 1.5 | NR Lite Net radiometer [a] |
| | D | $R_n$: $SW_{net}$,$LW_{net}$ | 1 | CNR2 Net radiometer [a] |
| | W | $L_vE, H, u*, z_0$ | 1 | |
| Eddy covariance fluxes | WS | $L_vE, H, u*, z_0$ | 1.5 | IRGASON [b] |
| | D | $L_vE, H, u*, z_0$ | 2 | |
| | W | $G, T_{soil}$ | -0.15;-0.10 to 0.2 | |
| Soil | WS | $G, T_{soil}$ | -0.05;-0.04 to 0.1 | T107 [b], HFP01SC [c], HFP01 [c] |
| | D | $G, T_{soil}$ | -0.05;-0.04 to 0.1 | |
| | W | $T, RH, U, WD, P$ | 2.5 | 107 T-prb [b];05108-45-L Wind[d];HPM155 T-RH prb[e] |
| Standard meteorology | WS | $T, RH, U, WD, P$ | 1.5 | 107 Temp. probe [a], IRGASON [a] |
| | W | $T, RH, U, WD, P$ | 2.5 | 107 Temp. probe [a], IRGASON [a] |
| b | | | | |
| Radiosonde profile | W | $T,RH,U,WD,P$ | 0-↑2000 | iMet-4 Radiosonde [f] |
| | D | $T,RH,U,WD,P$ | 0-↑2000 | |
| UAV profile | W | $T,RH,P$ | 0-500 | iMet-XQ2 UAV [f] |
| | D | $T,RH,P$ | 0-500 | |

[a]Kipp & Zonen, Delft, The Netherlands; [b]Campbell Sci., Logan, Utah, USA

[c]Hukseflux, Delft, The Netherlands; [d]Young Company, Traverse City, Michigan, US

[e]Vaisala, Helsinki, Finland;[f]InterMet Systems inc. Grand Rapids, Michigan



### 3.3 Airborne observations

We used two airborne instrument carriers: a radiosonde balloon and a UAV. These were equipped with similar sensor packages
that provided measurements of $T$, $RH$, $U$, $WD$ and $P$ for the radiosonde and $T$, $RH$ and $P$ for the UAV (details in Table 1b
and in Suárez et al. (2020). The radiosonde balloons were launched from two locations described in 3.1. At both locations, we
performed intensive campaigns on November $21^{st}$ over the water surface and November $22^{nd}$ over the desert surface (Fig. 2b),
where we launched balloons at 09:00, 12:00, 15:00, 18:00, and 21:00 local time (LT). Balloons typically reached an altitude
of 10 km and drifted away horizontally up to a distance of 50 km northeastward of their launching sites. Vertical profiles of $\theta$,
$q$, $U$ and $WD$ were obtained from the radiosonde to characterise and estimate the ABL height ($h$), using the surface pressure
level of the SDH ($\sim$650 hPa). This height was estimated through the maximum vertical gradient of $\theta$ (Sullivan et al., 1998).
The UAV was flown simultaneously from the same two locations as the balloon launches (described in 3.1) from the ground
to up to 500 m above the ground level (agl) from the surface on November $21^{st}$ and $22^{nd}$ every 30 min from 09:00 to 12:00
LT. From these flights we obtain the vertical profiles of $\theta$ to characterize the first 500 m agl of the ABL. UAV flights were,
unfortunately, not possible after 12:00 LT due to high winds.

### 3.4 WRF regional modeling

To complete the analysis of the E-DATA experiment, we reproduce the same period using the Weather Research and Forecast-
ing (WRF) model version 3.7 (Skamarock et al., 2008). We aim to study the atmospheric circulation that is formed daily from
the Pacific Ocean to the Andes western slope. We follow the methodology suggested by Jiménez et al. (2016), which consists
of performing consecutive, short WRF runs initialized at 0 UTC and running for 48 h. The first 24 h of each run is used as a
spin-up for the physical parameterizations and the 24-48 h to represent the weather conditions of the simulated day. Therefore,
we only analyzed and evaluated the period 24-28 hours. This methodology ensures that each simulated day starts with its real
respective initial and boundary conditions. Initial and boundary conditions are taken from ECMWF ERA-INTERIM reanalysis
data for 20°S /68°W with a 0.5° spatial resolution. By using this dataset input, every six-hours there is an update of the ten-
dencies due to the large-scale forcing. Figure 2c shows the horizontal distribution of the four two-way nested model domains,
detail information can be found in Table A1, Appendix A. The inner domain (D04) includes all the measurements gathered
in the E-DATA experiment. In the vertical direction, we imposed 61 non-equidistant grid following an exponential shape that
maximizes the number of vertical levels in the boundary layer, i.e. 40 within the first 2000 m. Several physical processes such
as radiation, surface and boundary layer, convection, microphysics, and land surface model are parameterized in WRF, they
are also detailed in Table A1. A comprehensive model validation from both surface and vertical variables is presented in detail
in Appendix A3.





## 4 Results and discussions

The comprehensive data-set of E-DATA enables us to study the main processes governing open water evaporation in arid conditions. The main factors under analysis are radiation, turbulent mixing and water vapour pressure deficit. In this section we systematically study how the local and regional scales contribute to the diurnal variability of E.

The results section is organized as follows. First, it shows the differences in E depending on where the measurements were taken: over water, wet-salt and desert surfaces (Section 4.1). Then, the main focus is on the results obtained at the water surface.

Additional local surface measurements and boundary layer profiles that help to define the distinct E regimes are presented in section 4.2. Finally, section 4.3 shows the WRF modeling results that help us to understand the local measurements of E in the saline lake by adding a regional perspective to the air flow.

### 4.1 Local measurements: Surface energy balance

Figure 3 displays the average diurnal cycles of the SEB terms, i.e. net radiation ($R_n$), ground ($G$), latent ($L_vE$), and sensible ($H$) heat fluxes observed above water, desert and wet-salt surfaces. All the sites are located within in a radius of 10 km. Typical daytime values of the SEB terms are summarized in Table 2.

Our measurements show exceptionally high $R_n$ levels over the water surface ($\sim$950 W m$^{-2}$), less for the desert surface ($\sim$700 W m$^{-2}$) and considerably less for the wet-salt ($\sim$500 W m$^{-2}$). The $R_n$ daily cycles follow a typical sinusoidal diurnal cycle

with the intermittent presence of high clouds (Fig. 3). In the absence of four-component radiation measurements at the three sites we cannot provide a detailed breakdown of the short and long-wave radiation components to $R_n$. Assuming that the incoming short- and long-wave radiation terms are equal for all sites, and taking the near surface soil temperature ($<$1 cm depth) as a proxy for the long-wave outgoing radiation we can see the following (see also Table 2). Maximum incoming shortwave radiation is $\sim$1250 W m$^{-2}$, which is close to the solar constant at the top of the atmosphere ($\sim$1360 W m$^{-2}$) probably due

to the high altitude and dry conditions of the study site. At $\sim$250 W m$^{-2}$ maximum long-wave incoming radiation is rather small, due to the thin atmosphere and mostly cloud-free conditions. The albedo of the desert surface is closer to the albedo of the water than the wet-salt (0.21 vs 0.12), but it is mainly the difference in surface temperature (27 vs 22°C) that leads to a larger long-wave outgoing radiation loss and thus lower $R_n$. Compared to water, the wet-salt surface has a comparable surface temperature (20 vs 22 °C), but it is the considerable difference in albedo (0.58 vs 0.12) that leads to a larger short-wave

outgoing radiation loss and thus much lower $R_n$.

While $R_n$ shows a clear sinusoidal diurnal cycle, the SEB heat fluxes show two distinct regimes. The first occurs in the morning (07:00 - 12:00 LT) and is characterized by very low values of $L_vE$ ($<$ 30 W m$^{-2}$), almost zero $H$, over the water surface for instance. As a result, most of the radiative available energy is used to heat up the lake water and underlying soil sediment ($G \approx R_n$, with values up to 600 W m$^{-2}$). The second regime occurs in the afternoon to early evening (12:00 - 20:00 LT). It

begins with a rapid (2-hour) rise in $L_vE$ and to lesser extent also in $H$ at the expense of $G$, which diminishes in the afternoon to the point at which it becomes negative and provides additional energy, in addition to the decreasing $R_n$, to the turbulent





**Figure 3.** Diurnal cycle of the surface energy balance (SEB) observed during the E-DATA field experiment. Mean separated components are shown in colour lines and maxima and minima by shadings. (a), (b) and (c) show the SEB over the water, wet-salt and desert surfaces, respectively. Vertical dotted lines indicate the time of regime change. A photograph of each SEB-station installed is shown at the right side of each graph.



**Table 2.** Radiation and surface energy balance variables measured and inferred from complementary measurements above water, wet-salt and desert surfaces during the E-DATA experiment. Maximum mean values of incoming short- ($S_i$) and long-($L_i$)wave radiation, albedo, surface temperature ($T_s$), $R_n$, $G$, $L_v E$, $H$ and daily mean of Bowen ratio.

|  | $S_i$ | $L_i$ | Albedo | $T_s$ | $R_n$ | $G$ | $L_v E$ | $H$ | Bowen ratio |
|---|---|---|---|---|---|---|---|---|---|
|  | $[Wm^{-2}]$ | $[Wm^{-2}]$ | [ - ] | [°C] | $[Wm^{-2}]$ | $[Wm^{-2}]$ | $[Wm^{-2}]$ | $[Wm^{-2}]$ | [ - ] |
| Water | 1250 | 250 | 0.12 | 22 | 950 | 500 | 500 | 100 | 0.2 |
| Wet-salt | 1250 | 250 | 0.58 | 20 | 500 | 400 | 50 | 200 | 4 |
| Desert | 1250 | 250 | 0.21 | 27 | 750 | 200 | 5 | 500 | 100 |

fluxes $H$ and $L_v E$. Focusing on $E$, its behaviour is atypical for surfaces where water is plentiful and E is mainly driven by the available energy (energy-limited system). Here, our analysis shows that in the morning E is very small even though the levels of $R_n$ are very high, which indicates that it is limited either by turbulence or $vpd$ (see Section 2). In turn, in the afternoon, the
E-regime changes to the typical $R_n$-limited type to the point at which it requires additional energy from the soil ($G$ becomes negative even before 15:00 LT).

On the wet-salt and desert surfaces, two similar surface flux regimes are observed, indicating that this feature dominates the entire study site and is not only specific to the water surface. However, there are interesting differences between the wet-salt and desert surfaces with respect to the water surface. In the wet-salt surface all the heat fluxes are much lower, reflecting the
limited amount of $R_n$ available (about half of that of water, as shown in Table 2). Furthermore, the roles of $H$ and $L_v E$ are reversed, i.e. it is $H$ that suddenly increases when the afternoon regime commences (water and wet-salt surfaces Bowen ratio water of 0.2 and 4, respectively). The salt-crust reduces the soil evaporation of the wet-salt surfaces, in addition to the salt lowering E in general (Kampf et al., 2005) (Fig. 3b). In the desert, $L_v E$ is zero all day and $R_n$ is balanced between $G$ and $H$ (Fig. 3c). The two regimes are clearly visible and show similarities to the wet-salt regime, with the difference that in the
morning regime $G$ and $H$ are similar while in the afternoon regime $H$ is dominant.

In the next section we further analyse the mechanisms that explain the two-regime behaviour in the local SEB fluxes and link them to a description of the local boundary-layer profiles.

## 4.2 Local perspectives: from surface to atmospheric boundary layer

Figure 4a shows the mean daily cycle of wind speed ($U$) and direction ($WD$) over the water surface. Over wet-salt and desert surfaces, a similar diurnal variability is observed (Appendix B). The morning regime with low turbulent fluxes is related to conditions of very low wind speed ($U < 1$ m s$^{-1}$) and variable wind direction between the S and SW. The afternoon regime with high turbulent fluxes is related to high wind speeds ($U > 10$ m s$^{-1}$) and a well-defined wind direction from the West. This





**Figure 4.** (a) Mean diurnal cycle of wind speed ($U$), turbulent kinetic energy (TKE), and wind direction ($WD$) of a representative day (November $18^{th}$), (b) mean diurnal cycle of aerodynamic resistance ($r_a$), (c) air temperature ($T$), surface temperature ($T_s$) and thermal gradient ($-dT$) and (d) air specific humidity ($q$), surface saturated specific humidity ($q_s$) and moisture gradient ($-dq$) observed over the water surface. Vertical dotted lines indicate the time of turbulent regime change, blue dashed lines the sunrise-sunset, and shadings represent maximum and minimum observations. Observations from November $15^{th}$-$24^{th}$ 2018.

wind pattern is typical of this season and has been observed regularly in 2015, 2016, and 2017 as well.

Figure 4b shows that as a result of the low wind speed in the morning the aerodynamic resistance is very high ($r_a > 400$ s m$^{-1}$; turbulent kinetic energy, TKE $\sim 0$ m$^2$ s$^{-2}$) and E in SDH can be regarded as turbulence-$vpd$-limited (see Equation





1). Note that in absence of any wind the water surface is extremely smooth (de la Fuente and Meruane, 2017; Suárez et al., 2020) and subsequently the surface roughness does not assist in generating shear. Additionally, $H$ over the water is nearly zero as well, meaning that the high $r_a$ is the result of the absence of both shear and buoyancy-generated turbulence. In contrast,

for the desert surface, this occurs when the winds are equally low but the temperature gradient is steep enough to sustain a mainly buoyancy-driven $H$ of about 200 W m$^{-2}$. In the afternoon, when the strong wind starts, $r_a$ drops dramatically and TKE increases in the same manner (4 m$^2$ s$^{-2}$, see inset Fig. 4a), which results in the onset of the fluxes, when the E regime goes from a turbulence-$vpd$-limited to a radiation-limited E regime.

We now connect the gradients of temperature (linked to buoyancy forced turbulence) and moisture (linked to the $vpd$) between

the surface of the water and the atmosphere at 1 m height, as well as how these affect E. Figure 4c shows the daily cycle of near-surface temperature ($T_s$), $\sim$1 m height air temperature ($T$) and surface-1 m thermal gradient ($dT$), over the water surface. The early morning (03:00 - 07:00 LT) displays low values of $dT$, where both air and water surface set below 0 °C and stay nearly -constants due to the formation of water ice. In the late morning, $T_s$ and $T$ increase rapidly, and mild thermal gradients corroborate the low $H$ (Fig. 3a) and no buoyancy-generated turbulence. In the afternoon, $dT$ increases to about 7 °C and then

falls in accordance with the available radiation. Note that there is a lag between $T_s$ and $T$ peaks, where $T$ decreases earlier than $T_s$. This behaviour is explained by the effect of the wind and cold air advection, which is stronger at 1 m than at the surface. The latter is corroborated by the $H > 0$ W m$^{-2}$ shown in Figure 3a from 12:00 LT.

Figure 4d shows the daily cycle of saturated-specific humidity ($q_s$), 1 m height specific humidity ($q$) and the surface-1 m humidity gradient ($dq$), over the water surface. In the morning, $dq$ are small due to $q_s$ being constant according to $T_s$ (ice on

water). Note that the gradient is taken between 1 m and $z_{0h}$ (very close to the surface); This does not seem to warrant the system being labeled $vpd$-limited. However, the absolute $q$ of the IRGA (sensors in Table 1) is sensitive to calibration issues, therefore we hypothesize that the gradient very close to the surface could have been smaller, to such a degree that the lack of turbulence results in a thin, water-saturated layer that prevents the creation of a gradient, and as a result leads to very small values E. During the late morning $q_s$ increases according to $T_s$ and $q$ shows a sudden drop of about 1 g kg$^{-1}$, just before

the change in the wind regime. Finally, during the afternoon, $q_s$ reaches its peak and then falls according to $T_s$. Likewise, $q$ increases by 2 g kg$^{-1}$ revealing, together with $T$, an advection of cold and slightly moister air into the study site. The advection of heat and moisture is discussed below in the vertical profile measurements and WRF modeling results.

These surface gradients are very dependent on the diurnal evolution of the ABL. Here, we present the vertical profiles at the water and desert surfaces as observed in the morning (Fig. 5). During the morning in the desert, the vertical structure of

potential temperature, $\theta$ and specific humidity, $q$ follows the evolution of a prototypical dry convective ABL (Figures 5a and 5b). The morning starts (09:00 LT profile) with a shallow unstable layer, corresponding to the unstable surface layer, followed by a stable layer until 1000 m agl. Driven by the surface sensible heat flux ($H$ = 100 W m$^{-2}$), the ABL rapidly develops into a deep, well-mixed ABL (12:00 LT profile) where the boundary layer is capped by an inversion at $h$=1800 m. The entrainment of dry, warm air from above the ABL supports its growth to 12:00 LT. On the basis of the high warming observed from 09:00

to 12:00 LT (Fig. 5a), we have estimated a non-local contribution of warm air close to 140 W m$^{-2}$.

Contrary to this, in the early morning over the water surface, we observe, for both the $\theta$ and $q$ profiles (Figures 5c and 5d, 09:00





**Figure 5.** Morning vertical profiles of potential temperature ($\theta$), and specific humidity ($q$) over the desert surface (a and b) 20.35° S - 69.90° W at 3931 m asl in November $22^{nd}$ 2018, and over water surface (c and d) 20.27° S - 68.88° W at 3790 m asl on November $21^{st}$ 2018.





LT profiles) a transition from a stable to a close to well-mixed profile. The stable profile at 09:00 LT is quantified in 0.026 K m$^{-1}$, and starts to decrease its stability to 0.016 K m$^{-1}$ at 10:00 LT, 0.00 1 K m$^{-1}$ at 11:00 LT, and reaching a well-mixed type profile at 12:00 LT ($\partial\theta/\partial t$ with >0.001 K m$^{-1}$). From 11:00 to 12:00 LT the $\theta$-profile shows an entire well-mixed boundary

layer higher than 500 m, which is probably attributable to the desert convective ABL that is dominant on the study site (Fig. 5a). In the absence of wind and significant heat fluxes in the morning, the ABL is not driven by surface processes and weak, local (meso-scale) flows are likely to be dominant. Figure 6 shows the time-series of a typical day of $U$, and $T$ of a westward, upslope transect of meteorological stations (see Figure 2). Here, in the early morning (03:00 - 06:00 LT) a WNW flow is visible, in which cold air accumulates at the lowest station. Figure 6 also shows that in the course of the morning the wind direction veers

180° to ESE. The night downslope and morning upslope circulations are indicative of a katabatic (early morning)-anabatic (late morning) circulation between the low-lying saline lake and the surrounding mountain ridges. The anabatic circulation interacting with the top of the boundary layer potentially exhibits return flow that leads to a compensated subsidence over the lake (Whiteman et al., 2004), which would explain the eroding of the stable ABL in the course of the morning (Fig. 5c), as well as the warming observed in Figure 5a. In section 4.3 we return to the observational evidence by combining it with the analysis

of the WRF results in order to determine the diurnal variability of these local circulations.

   Figures 7a and 7b depict the wind profiles for the entire day as measured at the desert site. These profiles are very similar to those measured over the water. We therefore assume them as being representative of the entire study site. In the morning the winds are weak (<2 m s$^{-1}$ ) coming from different directions through the height, similarly to the ones represented in Figure 4a. In the afternoon the westerly wind increases strongly all across the boundary layer but is especially concentrated in a shallow

jet near the surface (between 0∼250 m) with maximum wind speeds of ∼15 m s$^{-1}$ at z∼80 m. These observations confirm that the surface winds are coupled to boundary-layer dynamics, which in turn are determined by the regional circulation flows.

   Figure 7c shows the evolution of the ABL depth as determined from the $\theta$ profiles over the desert and water surface. After the strong convective growth in the morning (∼530 m h$^{-1}$), we observe that the boundary layer height decreases rapidly in the afternoon, from 1600 m agl at 12:00 LT to 750 m agl at 17:00 LT over the water and from 1800 m agl to 650 m agl over

the desert. We attribute this decrease to a change in the wind regime, which allows the entrance of air masses with different temperature, moisture, and stability (Figures 8a and 8b). The mixed ABL values at 15:00 LT are cooler (decrease of ∼55 K) and moister (increase of 3 g kg$^{-1}$) than those observed at 12:00 LT (Figures 5a and 5b). Although the advected air is moisty, compared to the desert conditions ($q$ ∼0.5 g kg$^{-1}$), it is still characterized by a very low specific humidity ($q$ ∼3 g kg$^{-1}$) considering the above-water conditions ($q_s$ >15 g kg$^{-1}$). Hence, these moisty air mass does not significantly contribute to

the E (see $vpd$ subterm in equation 1). Moreover, the ABL during the afternoon at the desert site is characterized by a strong inversion capping at ∼500 m above ground, in which at 18:00 LT $\theta$ jumps, $\Delta$4 K (Fig. 8a) and $q$ jumps $\Delta q$ 2 g kg$^{-1}$ (Fig. 8b). Likewise, the ABL formed in the afternoon (after regional flow arrival) over the water presents a higher inversion capping that the desert (750 m agl), but lower $\theta$ jumps, $\Delta$1 K (Fig. 8c), and higher $q$ jumps $\Delta q$ 3 g kg$^{-1}$ (Fig. 8d). Returning to the surface fluxes presented in Figure 3a, we can now identify two mechanisms that increase $H$ in the afternoon. The first is wind-enhanced

turbulence, which increases the mixing efficiency between the surface and the atmosphere. Second is advection of cool air that increases the $\theta$-gradient and the subsequent near-surface instability of the atmosphere. Based on the turbulent heat fluxes (Fig.



**Figure 6.** (a) Transect-averaged of wind direction ($WD$) and wind speed ($U$) the E-DATA on November $21^{st}$, 2018. (b) The air temperature of the MET-station transect shown in Fig. 2b on November $21^{st}$, 2018.





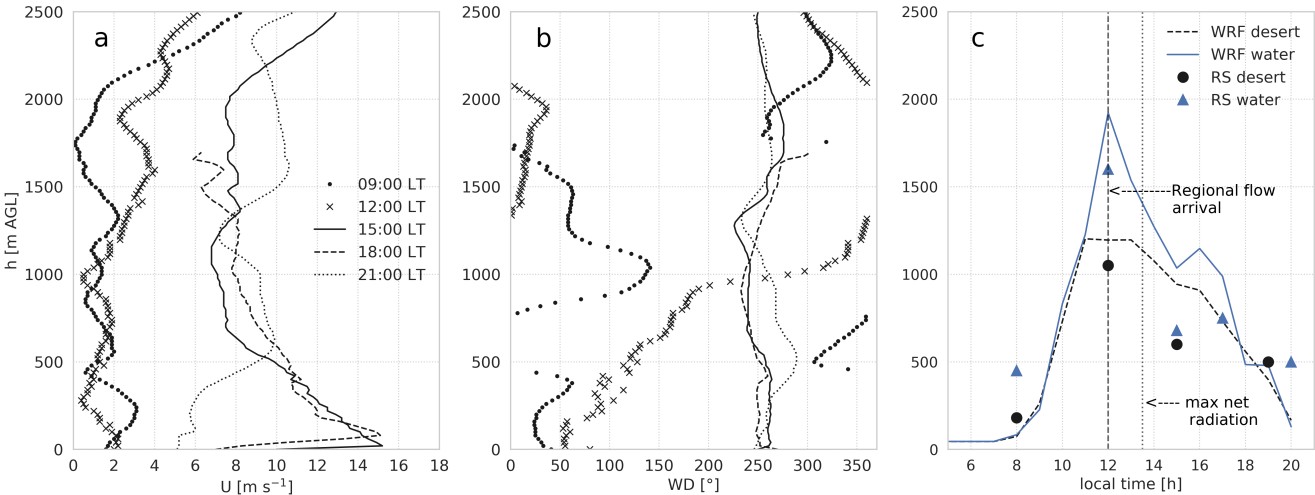

**Figure 7.** (a) Diurnal cycle of wind speed ($U$) and (b) wind direction ($WD$) vertical profiles and (c) boundary layer height ($h$) over the desert (November, $22^{nd}$; 20.35° S - 69.90° W; 3931 m asl) and water (November, $21^{st}$; 20.27° S - 68.88° W; 3790 m asl) surfaces determined by radiosounding (RS) and WRF simulation.

3) and the ABL height (Fig. 7b), and using the second term of Equations (3) and (4) as a residual, we quantify in Table 3) the local (surfaces fluxes), non-local (entrainment) and regional (advection) contributions to the mixed-layer tendencies of $\theta$ and $q$ between 15:00-18:00 LT. It is not surprising that with $L_vE$=0, the increase in humidity of 0.2 g kg$^{-1}$h$^{-1}$ is entirely accounted

for by regional advection (See computation details in Appendix C). Here the overall trend is relatively small (+0.33 K h$^{-1}$), given the relatively large $H$=400 W m$^{-2}$ (17:00 LT), due to the cool-air advection, which largely cancels the local heating.

  The afternoon profiles over water (Figs. 8c and 8d) show that on the arrival of the afternoon wind regime, the stably stratified

**Table 3.** The local (surface fluxes), non-local (entrainment flux) and regional (advection) contributions of $\partial q/\partial t$ and $\partial \theta/\partial t$ (equations (3) and (4)) corresponds to the period between 15:00 and 18:00 LT. These contributions are therefore averaged over this period and were taken above desert and water surfaces. Note that observations above the desert follow satisfactorily the assumptions of the Mixed-layer equation (3) and (4). Total tendencies, local and non-local contributions are based on SEB stations and radiosounding measurements, whereas advective contributions are estimated as a residual of equations (3) and (4) (see computation details in Appendix C).

| | $h$ | $\partial q/\partial t$ | Local | Non-local | Regional | $\partial \theta/\partial t$ | Local | Non-local | Regional |
|---|---|---|---|---|---|---|---|---|---|
| | [m] | [g kg$^{-1}$ h$^{-1}$] | [g kg$^{-1}$ h$^{-1}$] | [g kg$^{-1}$ h$^{-1}$] | [g kg$^{-1}$ h$^{-1}$] | [K h$^{-1}$] | [K h$^{-1}$] | [K h$^{-1}$] | [K h$^{-1}$] |
| Desert | 500 | 0.20 | 0.00 | 0.00 | 0.20 | 0.33 | 2.69 | 0.00 | -2.36 |
| Water | 680 | 0.28 | 0.86 | 0.01 | -0.59 | 0.33 | 0.66 | -0.0006 | -0.324 |

boundary layer up to 500 m present at the end of the morning (Figure 5c and 5d) becomes progressively eroded. In contrast



**Figure 8.** Afternoon vertical profiles of potential temperature ($\theta$), and specific humidity ($q$) over the desert surface (a and b) 20.35° S - 69.90° W at 3931 m asl on November $22^{nd}$ 2018, and over the water surface (c and d) 20.27° S - 68.88° W at 3790 m asl on November $21^{st}$ 2018.





to the eroding shallow mixed layer in the morning, in the afternoon, the destruction of the existing boundary layer structure is

driven by the surface processes. This process is explained by: a) enhanced mechanical turbulence from the strong winds of the near-surface jet (Fig. 7a); b) higher surface temperature (Fig. 4c) by the wind-induced mixing of the shallow water layer, and c) enhanced instability due to the cold air advection. This results in a shallow unstable layer, ranging from 0 m to ~150 m above the water surface between 15:00 LT and 18:00 LT. These levels are similar to the depth of the jet shown in Figure 7a. Regarding the moisture budget, the arrival of the wind flow in the afternoon moistens the unstable layer, while wind shear mixes it, result-

ing in steadily better-mixed q-profiles. At 21:00 LT an around 400 m-deep well-mixed boundary layer has developed over the water surface. Considering the budgets of local, and non-local versus regional contributions to the $q$ and $\theta$ mix layer tendencies over the water surface (Table 3), we quantify a major local $q$ contribution of about 0.86 g kg$^{-1}$h$^{-1}$ between 15:00 and 18:00 LT, and a small non-local contribution of 0.006 g kg h$^{-1}$. This moisture contribution exceeds the $q$-tendency observed, which can be only balanced by the negative regional contribution (-0.59 g kg$^{-1}$h$^{-1}$). The negative regional contribution of $q$ confirms

that even though the advected air is moist compared to the desert conditions, this is still dry for the water surface conditions. The $\theta$-tendency behaves similarly to that of $q$, whose local contribution of heat is equivalent to double the tendency value, but it is compensated for by cold regional flow (negative $\theta$ contribution).

### 4.3    Regional perspectives: modeling multi-scale mechanisms influencing E at SDH

In the previous sections, the measurement results indicate that E in the SDH is largely controlled by small-scale local circulations during the night and morning. This E pattern changes in the afternoon by the formation and arrival of regional meso-scale circulations. In order to better quantify how this circulation influences E at the SHD, we analyse WRF model results of the atmospheric conditions surrounding the SDH, using the regional-scale model WRF. We focus on two issues. The first concerns evaluating whether our measurements are influenced by small flows from katabatic-anabatic effects that dominate nighttime

and morning boundary layer in the absence of strong local or regional forcing. The second and more important one is the quantification of insights into the mechanism that generates the strong winds in the afternoon and produce the enhancement of E.

The local conditions that dominate the E in the SDH are analysed in Figure 9, which depicts the wind flow and temperature in the study site at 07:00 LT calculated with a grid resolution of 1 km, i.e. an effective resolution of approximately 3 km. The cir-

culation is characterized by a downward flow from the surrounding mountains ($z$ >4500 m asl) around the lowlands ($z$ ≈3800) where the saline lake is located, which tends to accelerate over pronounced slopes and closely follows the shape of the terrain. However, the lowest temperatures shown at the bottom of the valley in our observations (Fig. 6) are less clearly recognizable in the model (Fig 9a), where low temperatures occur in the surroundings of the lake. This down-slope flow is responsible for the stratified layers observed over the water surface at 09:00 LT (Fig. 5c). This katabatic flow progressively decreases in the course of the morning

of the morning whereas a transition from stable to well-mixed layer occurs above the water from 09:00 to 12:00 LT (Fig. 5c). Here, we observe two processes that are responsible for the local circulation and the low surface fluxes over the water during the morning (Fig. 3a). The first , between 09:00 - 10:00 LT, is an anabatic radial flow from the lake to its surroundings (Fig.







**Figure 9.** E-DATA period-averaged wind flow WRF simulation of domain D04 at 1km-resolution. (a) Surface $U$ and $T$ at 07:00 LT. (b) Surface $U$ and $T$ at 10:00 LT. (c) $U$ and vertical wind ($W$) at 13:00 LT. The black dot represents the saline lake.



9b). The second one, a downward flow produced by the interaction between the anabatic flow with the thermally driven wind during the morning-afternoon transition. This flow shown in Figure 9c produces a compensated subsidence (Whiteman et al.,

2004) at the western margin of the SDH valley, which explains the morning stratification over the lake shown in Figure 5c.

To characterise the recurrence of the wind pattern and its robustness at larger spatial scales, Figure 10 shows averages over 10 days (E-DATA period) of zonal wind speed ($U$), temperature ($T$), and specific humidity ($q$) in the morning (10:00 LT) and afternoon (16:00 LT) in a SW-NE vertical cross-section of the Andes mountains obtained by the WRF model. In the morning

we identify two main zones with clear $U$, $T$ and $q$ conditions. The first corresponds to the coast ($z < 1$ km) over the ocean (70.3° W), where the marine boundary layer (MBL) is characterized by low westerly winds of 2 m s$^{-1}$ (Fig. 10a), a thermal inversion capping at ~1 km height (Fig. 10b) and a quite well-mixed MBL with a moisture ranging between 7 and 10 g kg$^{-1}$ (Fig. 10c). The second zone corresponds to the western slope of the Andes (70.0° W to 68.5° W) above $z > 1$ km. This zone presents a very low $U$ (~1 m s$^{-1}$) that increases to 2 m s$^{-1}$ at the surface up-slope, producing a small local circulation in the

SDH basin (see red square in Fig. 10a). Likewise, there is a thermal contrast between the land and the top of the MBL (5 K) and incipient heating in the surface (70.0 ° W) together with a vertical thermal stratification of the atmosphere of 0.6 K per 100 m. Finally, low values of moisture are observed at middle altitude lands (~4 g kg$^{-1}$), with a variation ~ -1 g kg$^{-1}$ per km ascended on the slope (Fig. 10c).

During the afternoon, the morning conditions rapidly intensify. The $U$ increases at the surface >10 m s$^{-1}$) along the slope,

with a steep variation in its vertical profile, i.e., the weakest zonal winds are between 2 and 4 km asl (~1 m s$^{-1}$). Above ~4 km asl, typical synoptic southwesterly winds are found with speeds around 5 m s$^{-1}$ (Fig. 10d). The thermal contrast between the MBL top and the inland desert surface increases up to 10 K in the afternoon, in association with intense land warming. This strong wind circulation is characterized by higher values of the specific humidity (4 - 6 g kg$^{-1}$) from the top MBL ($z$: 1-2 km and longitude 70.3° W) along the side of the Andes slope (Fig. 10f). This strong advection follows two paths, one

reaches the SDH and increase the specific humidity from 1 to 3.5 g kg$^{-1}$, and the other one returns back westward at ~2 km asl. Two additional mechanisms on the western slope of the Andes that enhance the surface wind flow are also reproduced by the numerical experiment in WRF. The first mechanism is an anabatic flow formed at the midlands (70.0° W) driven by the high sensible heat fluxes, which corresponds to 73% of $R_n$. The second mechanism that is superimposed on the anabatic flow is a surface flow acceleration along the slope, which we recognise as flow channeling. This channeling is given by the shape of

the topography and the subsidence produced by the SE subtropical anticyclone (Rutllant et al., 2013) over the the SE Pacific Ocean and the western slope of the Andes . The flow is then channeled down into the SDH basin from the SW, producing local subsidence (Fig 9c). In summary, the origin of the strong wind that controls the evaporation in the Salar del Huasco originates in the regional daily atmospheric circulation from above the MBL to the Atacama Desert.



**Figure 10.** Vertical cross-section of diurnal atmospheric circulation on the Pacific Ocean Andes western slope, simulated with WRF for the E-DATA diurnal-averaged along 21.5° S. Black arrows represent $U$ (zonal winds) and $WD$ and the red square the SDH. (a), (b), and (c) represent $U$, $T$ and $q$, respectively, during the morning, and (d), (e) and (f) are the same for the afternoon.





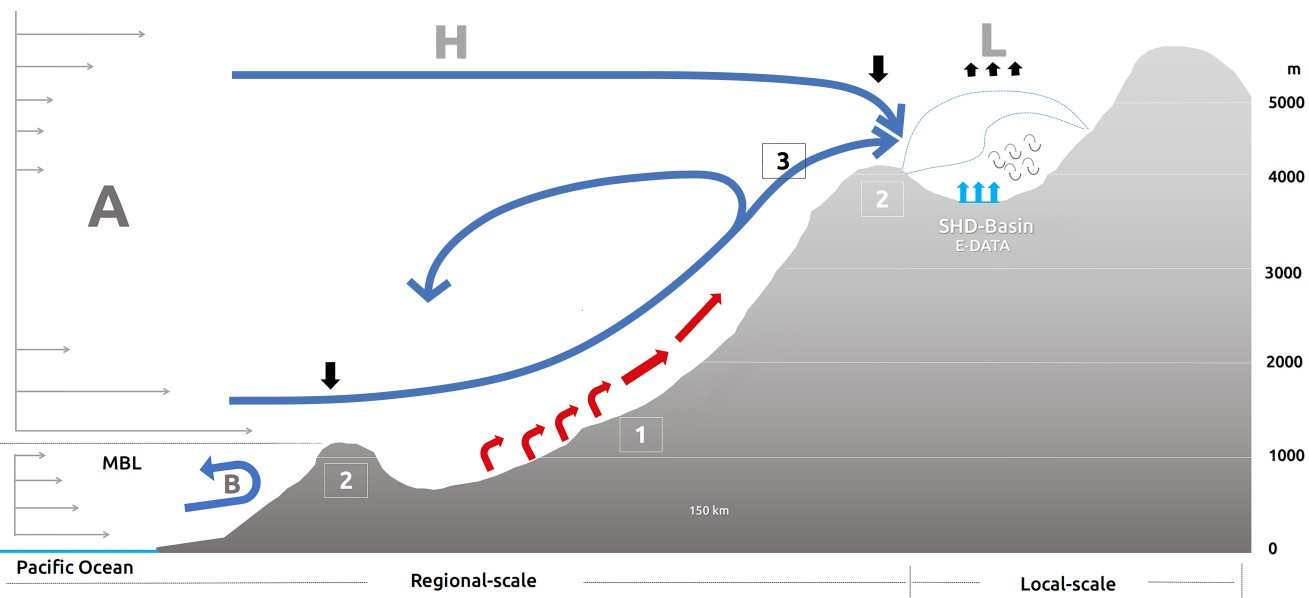

**Figure 11.** The regional and local circulation patterns that act between the Pacific Ocean and the western slope of the Andes. H and L (black arrows) represent the synoptic high and low pressures. A) corresponds to the regional zonal circulation from the top of MBL and (B) corresponds to the zonal circulation within the MBL resulting from the coastal daytime warming. Grey arrows (left) represent the regional zonal wind intensity. 1 (red arrows) indicates the anabatic flow. 2 schematises the topographic channelling process. 3 (blue arrows) shows the advection. Blue dotted line represents the formation of boundary layers, and cyan arrows the E produced by turbulence (circular arrows).

## 5    The role of atmospheric circulation on evaporation

Analysis of the observations carried out during the E-DATA field experiment and the WRF simulations enable us to propose a physically based explanation of the main role played by the wind in the control of evaporation in the Salar del Huasco basin. The overall dynamic of the regional atmospheric circulation interacting with local-scale processes described in the results is depicted in Figure 11. The regional circulation is a result of a multitude of processes and mechanisms interacting at scales from about 100 km to 100 m close to SDH. The main dominant atmospheric circulation driver is the thermal contrast between the above the MBL of the Pacific Ocean and the western slope of the Andes. Two principal and independent atmospheric circulations are dominant at daytime on the regional scale: (A) an thermally driven flow from the above the MBL (Rutllant et al., 2003) and (B) a local sea breeze formed within the MBL that interacts with the coastal mountains (Rutllant et al., 2003; Lobos Roco et al., 2018). Two other phenomena that occur on smaller spatial length scales reinforce this regional flow: anabatic flow (1) and topographic channeling (2), which enhance the inland flow from above the-MBL (A) when this reaches land. The interaction of these meso-scale (A and B) phenomena results in (3) the horizontal advection of air masses driven by the surface winds. Part of this advection transports cold and dry air into the Andes highlands basins, whereas the other part returns back into the midlands forming a small cell (Rutllant et al., 2003). This thermally driven flow interacts with the synoptic flow





(4-5 km asl) when both arrive to SDH. This multi-scale regional circulation influences the evaporation at SDH in two ways:
by producing mechanical turbulence and by transporting the cold and dry air above the water surface. Both processes lead to
an abrupt transition in the diurnal variability of the evaporative pattern over open water: from being almost zero during the
morning to large evaporation from the noon.

This regional circulation has already been well studied. Rutllant and Ulriksen (1979) and Rutllant et al. (2003) describe obser-
vations of the southwesterly atmospheric circulation for summer and winter at 250 km S of the SDH, as a consequence of the
diurnal ocean-land thermal differences. This result corresponds to the circulation system (A) depicted in Figure 11. This same
pattern as we found has been also reported in numerical experiments performed in November 2008 by Rutllant et al. (2013)
as well. Rutllant et al. (2003) also suggested that the atmospheric circulation (B) in Figure 11 can be coupled to the system A.
This occurs when the marine subsidence inversion is weak, allowing for the entrance of marine air masses to the desert. Our
numerical experiment also shows this interaction, from where the air that is advected towards the SDH starts ((3) in Fig. 11).
The values of $q$ and $T$ at the level of 1 km asl in our results agree with the vertical profiles observed by Muñoz et al. (2011) for
the Atacama coast, which are the same as we observe arriving during the afternoon at SDH. However, further research must
investigate the origin of the moist and cold air mass that arrives at SDH, in order to accept or discard the origin suggested by our
results. Likewise, Falvey and Garreaud (2005) describe the predominance of westerly winds from the free atmosphere towards
the Andes western slope ((A) in Fig. 11) during the summer at 500 km N of the SDH. The regional atmospheric circulation for
dry periods described by Falvey and Garreaud (2005) agrees with our results for the dry season. Finally, the surface regional
atmospheric circulation was also found by Muñoz et al. (2018) in their analysis of surface wind measurements all around the
Atacama Desert. Muñoz et al. (2018) reported predominant SW surface wind speeds below 5 m s$^{-1}$ during the morning, which
intensify to 15 m s$^{-1}$ during the afternoon. These results agree with our observations (Fig. 4a) and our numerical experiments
(Figs. 10a and 10d).
With respect to the wind-related mechanisms that control evaporation, our research extends to previous observational studies
performed under similar environmental conditions. The modeling results of De La Fuente and Niño (2010) show that $R_n$ is
mainly balanced by $L_v E$, which is driven by the afternoon wind during summer in Salar Punta Negra (500 km S of the SDH).
They also reported a similar $L_v E$ diurnal cycle, which is close to 0 W m$^{-2}$ in the morning and has a sudden enhancement
in the afternoon caused by changes in the pattern of winds (de la Fuente, 2014; de la Fuente and Meruane, 2017). This result
agrees with our description of morning-afternoon turbulent regimes shown in Figure 3a, but also regarding the diurnal cycle of
$r_a$ (Fig. 4b). In a different region, the relationship between wind and $L_v E$ has been also observed by de De Bruin et al. (2005)
over a crop field surrounded by a desert area in Idaho, USA. de De Bruin et al. (2005) observed that the advection of dry and
warm air from the surrounding desert, shows a negative $H$ in the SEB, resulting in ratios of $L_v E / R_n > 1$. However, our results
show a different pattern, since over the three different surfaces $L_v E / R_n$ is always lower than 1 and $H$ is positive (Fig. 3). On
the other hand, Tanny et al. (2008) describe the $L_v E$ diurnal cycle in a water reservoir in northern Israel. They estimated the
evaporation rates using several models, and validated their estimates by means of direct evaporation measurements performed
with an eddy covariance system. They concluded that a better agreement between measured and estimated E occurs for models
that represent better the wind diurnal cycle compared to those that consider the wind contribution to be constant. This agrees





with our observations as shown in Figures 4a and 4b.

Our findings related to ABL dynamic show different results above the SDH compared to classical interpretation of atmospheric boundary layers (Stull, 1988). However, they compare well with previous studies performed in different environments. First, the morning dynamic of the ABL described in section 4.3 has been also reported by Whiteman et al. (2004) and Whiteman (1989) in closed mountain basins in the Alps, the Rocky Mountains and Bush Creek Valley in the US. The authors show similar vertical profiles in the saline lake during the morning at the bottom of the valley. Moreover, the same dynamic of morning an-

abatic wind and the consequent compensated subsidence has been observed via a conceptual model by Whiteman et al. (2004). Our results share some similarities with those obtained by Batchvarova and Gryning (1998), describing changes in the boundary layer due to the sea-land breeze advection conditions in Athens. Even when geographical differences between these two locations exist, the profiles show the same diurnal evolution of the thermal structure observed over the water surface at SDH (Fig. 8c and 8d). Likewise, the wind vertical profile reported by Batchvarova and Gryning (1998) agrees with our observations

(Fig. 7a). This wind profile characterized by a surface jet has also been observed by Raynor et al. (1979) in the Atlantic's US coastal ridge under summer sea-land breeze conditions.

This research might be extended to contribute to the understanding of the climatology of the evaporation process. For instance, more work needs to be done to obtain evaporation estimates over different seasons, such as the summer rainy season over the desert, where synoptic and radiative conditions change completely. Similarly, more work is needed to reduce the uncertainties

in observations, for example by using a range of different methods to integrate the surface heterogeneity. Additionally, WRF simulations might enable us to design numerical experiments to improve our understanding of changes in the regional circulation that can affect wind patterns and therefore evaporation in the highlands. Our results demonstrate that there is significant variability in evaporation at scales below 1 km, and the relevance of coupling regional circulations to micrometeorological experimental studies, thus helping to improve the representation of E in models, and consequently, improving water management

in arid regions.

## 6 Conclusions

We investigate the diurnal variability of evaporation in a saline lake at high altitude. By combining surface and atmospheric high-resolution observations taken during the E-DATA field experiment and high-resolution WRF modeling results, we have

found that the wind, governed by thermal and orographic differences on different spatial scales, is the main driver of evaporation in the Salar del Huasco. The absence of turbulence (wind) in the morning produces a high aerodynamic resistance that inhibits the transport of moisture from a saturated surface over the water into the atmosphere. This occurs when $R_n$ is not a limiting process. During the afternoon the arrival of the regional flow triggers turbulent kinetic energy (4 m$^2$ s$^{-2}$ after midday) driven by the shear. This enhancement in the turbulent mixing is accompanied by the advection of cold and dry air that enhance

the evaporation.

More specifically, our results distinguished two regimes: (1) the morning local regime dominated by high net radiation and





ground heat flux, low wind speed ($<2$ m s$^{-1}$), a low surface-atmosphere moisture gradient ($\sim$3 g kg$^{-1}$) and an extremely low evaporation rate ($\sim$0 W m$^{-2}$). During this regime, the principal limiting driver of evaporation is the mechanical turbulence, in the absence of which the saturated specific humidity over the water is unable to mix with the dry atmosphere. Similarly, the

available net radiation is almost totally transferred to the soil, acting as a secondary factor in controlling evaporation. (2) The afternoon regional regime dominated by surface fluxes of latent and sensible heat flux, high wind speed ($>10$ m s$^{-1}$), a very high surface-atmosphere moisture gradient ($\sim$10 g kg$^{-1}$) and a sudden increase in evaporation over the water (500 W m$^{-2}$) This regime is no longer limited by wind (turbulence), instead, the decrease in net radiation, in the transition to the evening, characterizes the limiting factor. Similar regime patterns are observed over wet-salt and desert surfaces. However, the most

representative and sensitive variable is the sensible heat flux. For this reason, we conclude that these regimes are representative of the SDH basin and indicate the complexity of the land-atmosphere interaction due to large variations on sub-daily scales and the sub-kilometer surface heterogeneity.

The afternoon regional regime also has an impact on the development of the atmospheric boundary layer, particularly under the afternoon regime. The vertical profiles observations show the interruption of the convective boundary layer growth over

the desert. Over the water, an initial mixed layer about 180 m deep is formed in the early morning by katabatic winds. This mixed layer dynamically evolves into a stable layer in the late morning due to a local circulation that entrains warm air aloft, creating a stable stratified layer with thermal gradients of 0.02 K m$^{-1}$. The afternoon regional wind stops this stabilisation and leads to the formation of an unstable layer driven by high levels of mechanical turbulence production ($u* \sim$0.65 m s$^{-1}$). Our explanation relates the local evaporation with regional atmospheric circulations. We found that the regional circulation is due

to three interconnected atmospheric phenomena occurring at different spatial scales: (i) at 4000 m the above of the MBL of the Pacific Ocean characterized by a strong flow towards the land (15 m s$^{-1}$), (ii) an anabatic circulation driven by the contrast land-ocean (10 K) and (iii) a channeling of the flow occurring at 3000 m. The concatenation of these three phenomena leads to the daily appearance of strong winds, which then enhances the mechanical turbulence and, therefore, evaporation. Our findings indicate the need to combine complete local measurements with regional modeling to understand the interactions of arid land

conditions conditioned by a cold Ocean and complex land topography.

*Data availability.* https://data.mendeley.com/datasets/c5s6zk2rmz/2T

**Appendix A: Uncertainty of observations and modeling details**

In this section we briefly address the uncertainties related to surface and airborne measurements performed during the E-DATA

field experiment and the WRF modeling results. Complementary information can be found in Suárez et al. (2020).





## A1 Surface observations

The Eddy Covariance (EC) method is regarded as the most reliable method to measure $L_vE$ and $H$ fluxes. However, energy balance non-closure (Eder et al., 2014; Mauder et al., 2007) is found everywhere. Our results show imbalances of the SEB that range between 15 and 30% (Suárez et al., 2020), which agrees with several field experiments performed in the last decades (Eder et al., 2014). In addition, some instrumental issues might contribute to measurement uncertainties related to the following three reasons: (1) to obtain $R_n$ over the desert and wet-salt surfaces we used a less accurate sensor that did not measure all four radiation components, as opposed to that used at the water surface. For that reason, $R_n$ over the desert was corrected (see section 3.1) due to the unrealistic values we obtained. However, $R_n$ measurements might still mean an overestimation of the wet-salt surfaces, which would contribute to energy balance closure problems; (2), $G$ was measured using soil flux plates buried 5 cm from the air-surface interface (desert, water, and wet-salt surfaces). Consequently, $G$ must be corrected to account for heat storage in the soil or in the water body. The different surfaces complicated the installation of the soil sensors, which might underestimate $G$, which is an important component of the SEB at the SDH (de la Fuente and Meruane, 2017); (3) The exchange processes on larger scales might have a significant influence on the energy balance, due to the landscape heterogeneity (Foken, 2008).

## A2 Airborne observations

The uncertainty of the airborne measurements is related to the sensors carried by the radiosonde and UAV, the measurement footprints and to the disturbance the UAV's propellers might have caused to the sensor. Firstly, the sensors carried by the radiosonde and UAV were different models from the same manufacturer (Table 1), which might have led to differences in the observations. Secondly, the flight path (measurement footprint) followed by the two instruments was not exactly the same, in that the radiosonde flew at a height of around 10 km and up to 50 km NE of the launch site, while the UAV flew at an altitude of only 500 m from the launch site, with no horizontal travel. This means that different measurements of the vertical air column were made, which contributed to the uncertainty. Finally, to avoid the UAV's propellers disturbing the sensor during the take-off, we only use the profiles obtained during UAV landing, i.e. from 500 m to ground level. Nevertheless, during the landing, the propellers also might affect, although to a lesser extent, the sensor readings.

## A3 WRF modeling results

The following subsection includes detailed information of WRF numerical settings in Table A1. Moreover, this section includes the validation of WRF variables with surface and vertical observations, shown in Figure A1.

Table A1 describes the numerical settings of the model for input files, time control, domains, physics schemes, and dynamics. The initial and boundary conditions are obtained from ECMWF ERA-INTERIM reanalysis data for 20°S / 68°W with a spatial resolution of 0.5 degrees, that includes a six-hours update of the tendencies, due to the large-scale forcing. We modelled the





**Table A1.** Numerical settings used in WRF simulations organized by nested domains.

| Time control | | | | |
|---|---|---|---|---|
| Starting date | $13^{th}$, November 2018 | | | |
| Ending date | $24^{th}$, November 2018 | | | |
| *Domains* | D01 | D02 | D03 | D04 |
| Time step | 50 s | | | |
| dx | 27 km | 9 km | 3 km | 1 km |
| dy | 27 km | 9 km | 3 km | 1 km |
| Vertical levels | 61 | | | |
| Top of the model | 15790 m ($10000\ p_a$) | | | |
| *Physical parameterizations* | | | | |
| Surface layer | Monin-Obukhov scheme | | | |
| Radiation | RRTMG | | | |
| Boundary layer | YSU | | | |
| Land Surface | Unified Noah LSM | | | |
| Microphysics | WSM-3 class simple ice scheme | | | |
| Convection | Kain-Fritsch (new eta) | | | |
| *Dynamics* | | | | |
| Wave damping | yes | | | |
| Damping option | w Rayleigh | | | |
| z damp | 7000 m | | | |
| Damp coefficient | 0.2 | | | |
| Two-way nested | no | yes | yes | yes |
| Non-hydrostatic | yes | | | |

entire period of E-DATA, from $13^{th}$ to $24^{th}$ November 2018. As for the spatial domains (Figure 2c), the horizontal distribution

includes four two-way nested domains, in which the grid sizes are respectively 27 km for domain D01, 9 km for domain D02, 3 km for domain D03, and 1 km for the D04 inner domain. The D04 domain closely surrounds the study area. In its vertical direction, we defined 61 levels in an exponential fashion from the surface (including topography) to 15790 m of height, grouping 40 levels in the first 2 km. The physical processes represented are: RRTMG model for radiation physics (Iacono et al., 2008), for surface layer the Monin-Obukhov scheme (Janjić, 1996), the YSU scheme for boundary layer physics (Hong et al.,

2006), the Unified Noah Land-surface Model for land-surface physics (Ek et al., 2003), the WSM 3-Class simple ice scheme for microphysics, and the Kain-Fritsch (Kain and Fritsch, 1993) for convective scheme. Additionally, we adjusted the land-use map in order to set the saline lake in domains D03 and D04. Moreover, we also increased the SST 2 K in the WRF inputs of domains D01 and D03 according to the public information of the National Ocean and Atmospheric Administration (NOAA) from US department of commerce. We used additional special dynamic parameters within the model to filter the effect of the





unrealistic gravity waves caused by the strong topography of the Andes. The parameter we used has been the Rayleigh damping layer at 7000 m with a damp coefficient of 0.2 (Klemp et al., 2008).

The results obtained for domain D04 of the WRF model were validated by surface observation of CEAZA-met station (20.2°S - 68.8°W), which is permanently in operation since 2015. The first row of Figure A1 shows the validation of WRF variables $U$, $T$, and $q$, for an average period of E-DATA ($13^{th}$-$24^{th}$, November 2018). The best agreement is during the daytime when

evaporation occurs. We also validate our simulation using a station at the Pacific Ocean shore, Diego Aracena Iquique Airport station (20.5°S - 70.1°W), shown in the second row of Figure A1. This, aiming to validate our results obtained in domain D02, used for characterizing the regional circulation in section 4.3. We observe a good agreement in temperature and a slight overestimation in specific humidity. However, the model follows satisfactorily the diurnal cycle of observations. The radiosoundings launched over the desert site are compared with vertical profiles of WRF shown in the third row of Figure A1. We observe a

good agreement in $U$, $\theta$ and $q$ at noon and a good representation of the boundary layer height. Finally, based on our comparison of the wind speed during November 2015, 2016 and 2017, we conclude that our results for November 2018 are representative of the season climatology of the Salar del Huasco, since the wind pattern is very similar during the four years.

**Appendix B:  Desert and wet-salt wind, temperature and moisture conditions**

Similar to Figure 4 in the main text, Figures B1 and B2 show the mean diurnal cycle of wind speed and direction, aerodynamic

resistance, thermal and moisture gradients between the surface and the measurement level for desert and wet-salt surfaces. These figures support the homogeneous wind conditions (a, b) in the SDH-basin and contextualize the heterogeneous thermal (c) and moisture (d) gradients between the surface and the measurement height.

**Appendix C:  Explanation of the method used for quantifying local (surface), non-local (entrainment), and regional**

**(advective) contributions to the tendency term $\partial q/\partial t$ and $\partial\theta/\partial t$ in Table 3**

According to Equations (3) and (4), the tendency terms $\partial q/\partial t$ or $\partial\theta/\partial t$ shown in Table 3 represent the change of potential temperature and specific humidity within the boundary layer during a specified time period. These tendencies are calculated as the average of well-mixed values of $\theta$ and $q$ taken by the radiosoundings launched at 15:00 and 18:00 LT. For instance, over the desert site, Figure 8a indicates a  difference of ~1 K between 15:00 and 18:00 LT, i.e. a tendency term is 0.33 K per hour.

The local contribution corresponds to the turbulent fluxes, $\overline{w'q'_s}$ ($L_v E$) and $\overline{w'\theta'_s}$ ($H$) in the right-hand side of equations (3) and (4). This contribution is calculated using the averages surface fluxes (Fig. 3) and the averaged boundary layer height (Fig. 8) over the same time period. For example, for the desert site we measured ~0 W m$^{-2}$ of latent heat flux between 15:00 and 18:00 LT at a height of 500 m. The non-local contribution corresponds to the entrainment flux, $\overline{w'q'_e}$ and $\overline{w'\theta'_e}$ in the right-hand side of equations (3) and (4). This is calculated by using the vertical velocity obtained from the boundary layer growth

and time ($\Delta h/\Delta t$), and the change in the maximum vertical gradient, $\Delta q$ or $\Delta\theta$ between the same time period. Following



**Figure A1.** First row: diurnal average ($13^{th}$-$24^{th}$, November 2018) of 2-m $U$, $T$, and $q$ of WRF domain D04 and CEAZA-met station. Second row: diurnal average ($13^{th}$-$24^{th}$, November 2018) of 2-m $T$ and $q$ of WRF domain D02 and Diego Aracena airport-met station. Vertical profiles of $U$, $\theta$, and $q$ of WRF domain D04 and radiosounding launched during the E-DATA on $22^{nd}$ November 2018 at 12:00 LT over the desert.

**Figure B1.** (a) Mean diurnal cycle of wind speed ($U$) and wind direction ($WD$) of a representative day (November $18^{th}$), (b) mean diurnal cycle of aerodynamic resistance ($r_a$), (c) air temperature ($T$), surface temperature ($T_s$) and thermal gradient ($-dT$) and (d) air specific humidity ($q$), surface saturated specific humidity ($q_s$) and moisture gradient ($-dq$) observed over the **desert** surface. Vertical dotted lines indicate time of turbulent regime change and shadings represent maximum and minimum observations. Observations from November $15^{th}$-$24^{th}$ 2018.



**Figure B2.** (a) Mean diurnal cycle of wind speed ($U$) and wind direction ($WD$) of a representative day (November $18^{th}$), (b) mean diurnal cycle of aerodynamic resistance ($r_a$), (c) air temperature ($T$), surface temperature ($T_s$) and thermal gradient ($-dT$) and (d) air specific humidity ($q$), surface saturated specific humidity ($q_s$) and moisture gradient ($-dq$) observed over the **wet-salt** surface. Vertical dotted lines indicate time of turbulent regime change and shadings represent maximum and minimum observations. Observations from November $15^{th}$-$24^{th}$ 2018.



the example of the desert, between 15:00 and 18:00 LT ($\Delta h/\Delta t$) presents a small change and $\Delta q$ has almost no change, then no entrainment contribution was considered. Finally, the regional contribution corresponding to the larger-scale circulation quantifies the mean horizontal wind and the horizontal gradient of $\theta$ and $q$. This is the second term of the right-hand side of equation (3) and (4). In the absence of observations of the horizontal gradients and aiming to characterize the contribution using exclusively the observations gathered in E-DATA, the regional advection is estimated as a residual of each equation. Following the example above, at the desert surface between 15:00 and 18:00 LT there is no turbulent fluxes neither entrainment fluxes to $\partial q/\partial t$ because the latent heat flux is ~0 W m$^{-2}$ and $\Delta q$ is constant. However, the tendency term is 0.2 g kg$^{-1}$ per hour. This means that according to the budget equation (4), the only way to have a positive tendency of moisture is through the larger-scale advection.

*Author contributions.* The article was written by F. Lobos-Roco with the assistance of O. Hartogensis, J. Vilà-Guerau de Arellano and F. Suárez. The data were analyzed by F. Lobos-Roco and O. Hartogensis, who also contributed mostly to data processing. All data used in this study were gathered in a field experiment organized by F. Lobos-Roco, O. Hartogensis, F. Suárez and A. de la Fuente. F. Suarez and A. de la Fuente were responsible of funding of this field experiment (through projects ANID/FONDECYT/1170850 and ANID/FONDECYT/1181222). Data interpretation of local and regional atmospheric processes was assisted by J. Rutllant and R. Muñoz. All the authors contributed to the revision of the manuscript.

*Competing interests.* The authors declare that they have no conflict of interest

*Acknowledgements.* This research received financial support from the Chilean National Commission of Science and Technology through the projects CONICYT/FONDECYT/1170850 and CONICYT/FONDECYT/1181222. Support for Felipe Lobos was provided by the Wageningen University Ph.D. Sandwich Project no.: 5160957644. F. Suárez acknowledges support from the Centro de Desarrollo Urbano Sustentable (CEDEUS - CONICYT/FONDAP/15110020) and from the Centro de Excelencia en Geotermia de los Andes (CEGA - CONICYT/FONDAP/15090013). We thank Pedro Luca's family for their support during the fieldwork on their land. Finally, we acknowledge the reviewers Hugh Allen, and the two anonymous reviewers by their valuables contributions to this manuscript.





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
