# Peer review of "Local evaporation controlled by regional atmospheric circulation in the Altiplano of the Atacama Desert"

_Atmospheric Chemistry and Physics, 2020_

## Referee Comment (RC1)

This work presented at ACP deals with observations over heterogeneous surface (obtained during the E-DATA Experiment as well as airborne observations) and with numerical simulations run with the WRF model. The authors investigate the diurnal variability of evaporation over 3 different surfaces (water, wet-salt and desert) in the Altiplano of the Atacama Desert. The different processes and scales (regional and local mainly) controlling evaporation in arid regions, where some water environments can be present due to the large surface heterogeneity, are very relevant to understand the SEB at these zones of the Earth surface. The field experiment provides interesting data in order to answer the research question posed about the wind-induced turbulence in controlling the cycle of evaporation. With regards to the WRF simulations, in my opinion an important effort has done to get high vertical resolution simulations. I would like to underline the discussion on the influence of the different scales and physical processes on the evaporation rate at this site, which is really stimulating and well developed. I find the paper very interesting and well discussed and written, and I think it deserves to be published in ACP. Below there are some comments in order to improve the final version of the paper:

- It would be desirable than the authors discussed in a deeper way the uncertainty related to the SEB closure, especially those points related to advection and interaction between the local and regional scales. I know that this is quite a hard point to answer, but due to the open problem representing this SEB closure (or non-closure) it is necessary to face. This can be done in Appendix A, although the non-closure is not only a problem of uncertainty of observations. For example, the turbulence term in the equation (1) is usually a local term produced by local turbulence. How non-local turbulence produced by entrainment or advection can be considered in the evaporation rate?
- ERA-INTERIM from ECMWF is used for initial and boundary conditions (0.5° spatial resolution). Have you done any sensitivity test to use other source for initial and boundary conditions, as for example the NCEP-FNL data?
- I do not clearly find the average time used to evaluate the turbulent fluxes or parameters from the EC method. Have you done any sensitivity test to use different average times? This can be especially important for stable conditions at night (SBL) when using averaging times larger than 5 minutes can produce an overestimation of turbulent fluxes contaminated by sub-mesoscale (non-turbulent scales).
- The values of ground heat fluxes (G) showed on Table 2 are really large. I am surprised with these values. As you say in the manuscript these values are not the measured values by the instrument, but corrected by the storage term. I would like you to give more details about the way to evaluate the storage term and the value of this storage compared to G measured by the instrument, as important uncertainties can be in the evaluation of the storage term.

---

## Referee Comment (RC2)

Review of Local evaporation controlled by regional atmospheric circulation in the Altiplano of the Atacama Desert by Lobos-Roco et al. submitted to Atmospheric Chemistry and Physics Discussions.

General remarks

This manuscript provides a detailed analysis of the drivers of evaporation over the Salar del Huasco in the Altiplano of the Atacama Desert, a very shallow saline lake wherein the water within a confined catchment concentrates. Peak evaporation rates reduce the surface of the lake by 75% in only two months. The authors aim to unravel the role of local and regional scale processes by combining observations and detailed numerical model using the WRF model to describe moisture and energy budgets and the interaction of regional circulation and the boundary layer. The manuscript specifically contributes to a better understanding of the strong diurnal variability in evaporation over the lake, from nearly zero in the morning, to large fluxes from noon onward, driven by wind generated by processes governed by thermal and orographic differences at larger spatial scales. The contrast between the water, wet-salt, and dessert surfaces is studied in detail. I am specifically intrigued by the local scale effects of albedo. The authors thoroughly lay out the processes behind the distinct regimes, where despite Rn being very high, E is limited by turbulence in the morning, while E fluxes are high in the afternoon and limited by Rn.

The manuscript is logically organized, the research questions clearly explained, figures are clear, and results and discussion presented with minutious precision. I recommend the manuscript for publication in ACP after very minor revisions.

Specific remarks

I would rephrase the last sentence of the abstract to something like 'Our research contributes to untangling and linking local and regional scale processes driving evaporation across confined salt lakes in arid regions'.

Could the authors specify the overall depths of the lake in the introduction? Is 15 cm at the SEB station representative?

I suggest a few small textural corrections below.

I suggest the authors to write vpd as VPD.

L9 delete second and third 'the'

L19 replace 'and thus' with 'they'

L44 insert 'the' before 'diurnal'

L81 replace 'referred' with 'referring'

L307 replace moisty with moist

L467 insert layer after surface

L474 insert 'air at a' before 'saturated specific humidity'

---

## Author Comment (AC1)

This work presented at ACP deals with observations over heterogeneous surface (obtained during the E-DATA Experiment as well as airborne observations) and with numerical simulations run with the WRF model. The authors investigate the diurnal variability of evaporation over 3 different surfaces (water, wet-salt and desert) in the Altiplano of the Atacama Desert. The different processes and scales (regional and local mainly) controlling evaporation in arid regions, where some water environments can be present due to the large surface heterogeneity, are very relevant to understand the SEB at these zones of the Earth surface. The field experiment provides interesting data in order to answer the research question posed about the wind-induced turbulence in controlling the cycle of evaporation. With regards to the WRF simulations, in my opinion an important effort has done to get high vertical resolution simulations. I would like to underline the discussion on the influence of the different scales and physical processes on the evaporation rate at this site, which is really stimulating and well developed. I find the paper very interesting and well discussed and written, and I think it deserves to be published in ACP. Below there are some comments in order to improve the final version of the paper:

We thank the reviewer for his/her revision, which is helpful to improve the final version of this paper. Below in blue font, we provide answers to each reviewer's comment.

It would be desirable than the authors discussed in a deeper way the uncertainty related to the SEB closure, especially those points related to advection and interaction between the local and regional scales. I know that this is quite a hard point to answer, but due to the open problem representing this SEB closure (or non-closure) it is necessary to face. This can be done in Appendix A, although the non-closure is not only a problem of uncertainty of observations. For example, the turbulence term in the equation (1) is usually a local term produced by local turbulence. How non-local turbulence produced by entrainment or advection can be considered in the evaporation rate?

Thanks for your comments regarding the SEB non-closure. We are aware of the uncertainties related to it. Indeed, we briefly mention it in Appendix A1 in relation to the errors in the $R_n$ observations over the desert and wet-salt surfaces and G measurements over the water. We made a deliberate choice not to extend this discussion to a full treatment of non-local effects on the SEB closure uncertainty as it this is a (big) topic in itself and it is beyond the scope of this paper.

We do show the effect that non-local influences have on the humidity tendency (Table 3) which we split up in a local-, non-local- (entrainment), and regional (advection) contributions. These will influence the SEB fluxes, especially through the regional advection that enhances the local *vpd*.

We use equation (1) as an anchor to pinpoint the relative effect of processes steering $L_vE$, not so much as a model equation. The turbulence term therein, $r_a$, is estimated using EC fluxes which cannot distinguish between local and regional influences. One of the main messages of the paper is that the lack of turbulence limits $L_vE$ in the morning followed by a strong turbulent regime in the afternoon, brought about by a regional circulation, that rapidly enhances the $L_vE$ transport. From this it is self-evident that non-local govern the SEB and will impact the SEB (non)closure.

> We will add the following lines after line 515, in Appendix A1:

> "Advection and entrainment phenomena might add uncertainty to the SEB balance. However, our measurements limit us to evaluated them properly, and they are beyond the scope of this study."

ERA-INTERIM from ECMWF is used for initial and boundary conditions (0.5° spatial resolution). Have you done any sensitivity test to use other source for initial and boundary conditions, as for example the NCEP-FNL data?

> We did not perform a sensitivity analysis of a different dataset since the ERA-INTERIM data provided initial and boundary conditions that yield results that compared satisfactorily with the observations. We have added in appendix A2 the following sentence at line 533: "No additional data source were analyzed due to the high agreement of the WRF results based on ERA-INTERIM data sources and surface observations".

I do not clearly find the average time used to evaluate the turbulent fluxes or parameters from the EC method. Have you done any sensitivity test to use different average times? This can be especially important for stable conditions at night (SBL) when using averaging times larger than 5 minutes can produce an overestimation of turbulent fluxes contaminated by sub-mesoscale (non-turbulent scales).

> We processed fluxes both for 10- and 30min intervals and presented the 10min fluxes in the paper. Fluxes at both time intervals are very similar in magnitude and evolution. We did not consider 5min flux intervals, as we hardly experience stable conditions. We therefore do not expect the influence of sub-mesoscale contamination.

> The heat flux at night time is small and positive (typically 0 $W/m^2$ < H < 10 $W/m^2$). We discuss in Section 4.2 that the lack of transport is due to a very shallow boundary layer under the influence subsidence brought about by drainage of cold air from the surrounding mountain ridge. The flow associated with the drainage is too weak to create significant turbulence. Given the fact the we don't deal with stable conditions and that fluxes are as small as they are, we suggest to leave it with this internal discussion.

> We did update manuscript to incorporate the flux averaging interval Section 3.1 at line 145: "We used the EddyPro v 6.2.2 flux-software package (Fratini and Mauder, 2014) to calculate latent heat ($L_vE$), sensible heat (H) and friction velocity (u*), at 10-min averaging intervals."

The values of ground heat fluxes (G) showed on Table 2 are really large. I am surprised with these values. As you say in the manuscript these values are not the measured values by the instrument, but corrected by the storage term. I would like you to give more details about the way to evaluate the storage term and the value of this storage compared to G measured by the instrument, as important uncertainties can be in the evaluation of the storage term.

> To clarify, the values of G were measured at nominally 5 cm below the surface (desert and wet-salt) or ~1 cm in the sediment (water) using soil heat flux plates (SHP). These measurements are indicated in Table 1. The flux measurements were corrected for the heat storage above the plates according to Kimball and Jackson (1975). This procedure is explained in section 3.1, lines 150-154. In particular for the water surface, the heat storage was determined by integrating over a profile of 4 thermometers over the ~10cm water layer.

---

## Author Comment (AC2)

General remarks

This manuscript provides a detailed analysis of the drivers of evaporation over the Salar del Huasco in the Altiplano of the Atacama Desert, a very shallow saline lake wherein the water within a confined catchment concentrates. Peak evaporation rates reduce the surface of the lake by 75% in only two months. The authors aim to unravel the role of local and regional scale processes by combining observations and detailed numerical model using the WRF model to describe moisture and energy budgets and the interaction of regional circulation and the boundary layer. The manuscript specifically contributes to a better understanding of the strong diurnal variability in evaporation over the lake, from nearly zero in the morning, to large fluxes from noon onward, driven by wind generated by processes governed by thermal and orographic differences at larger spatial scales. The contrast between the water, wet-salt, and dessert surfaces is studied in detail. I am specifically intrigued by the local scale effects of albedo. The authors thoroughly lay out the processes behind the distinct regimes, where despite Rn being very high, E is limited by turbulence in the morning, while E fluxes are high in the afternoon and limited by Rn.

The manuscript is logically organized, the research questions clearly explained, figures are clear, and results and discussion presented with minutious precision. I recommend the manuscript for publication in ACP after very minor revisions.

> We thank the reviewer for the kind words of appreciation of our work and we are glad that he/she clearly recognized the main messages we wanted to convey with this paper. The reviewer's comments to improve the paper were limited. Below, in blue font, we provide answers to each of the reviewer's comments.

**Specific remarks**

I would rephrase the last sentence of the abstract to something like 'Our research contributes to untangling and linking local and regional scale processes driving evaporation across confined salt lakes in arid regions'.

> We agree with the reviewer's suggestion to rephrase this sentence. We will, however, change the word "salt" to "saline" to be consistent with the rest of the manuscript.

Could the authors specify the overall depths of the lake in the introduction? Is 15 cm at the SEB station representative?

> We think that the 15-cm depth at the SEB station represents the entire saline lake since we observed that the open water body presents mostly a homogeneous and flat bottom. We have added the lake depth in the introduction, line 23 "we focus on a particular, ~15-cm deep saline lake".

**Small textural corrections**

All small textural corrections mentioned below have been changed as suggested in the revised manuscript:

I suggest the authors to write vpd as VPD.
L9 delete second and third 'the'
L19 replace ' and thus ' with 'they'
L44 insert 'the' before 'diurnal'
L81 replace 'referred' with 'referring'
L307 replace moisty with moist
L467 insert layer after surface
L474 insert 'air at a' before 'saturated specific humidity'